# Machine learning-based prediction of metabolic dysfunction-associated steatotic liver disease using National Health and Nutrition Examination Survey (NHANES) data

Yong Zhang[1]❂, Xiang Liu[2]❂, Xingqiang Zhang[1], Yangfan Fei[3], Xiaoxu Li[1]*

1 Information Department, Meishan City People's Hospital, Meishan, Sichuan, China, 2 Information Technology Center, West China Hospital, Sichuan University, Chengdu, Sichuan, China, 3 Endocrinology Department, Meishan City People's Hospital, Meishan, Sichuan, China

❂ Co-first author, These authors contributed equally to the study.
* 38936174@qq.com

## Abstract

### Objective

With the global increase in obesity rates and lifestyle changes, metabolic dysfunction-associated steatotic liver disease (MASLD) has become a prevalent chronic liver disorder, affecting approximately 25% of the global population. This disease can progress to cirrhosis and liver cancer, posing a significant threat to public health. To facilitate early diagnosis and intervention, this study aims to develop an efficient and reliable prediction model for MASLD using machine learning algorithm.

### Methods

This study included 9,232 participants aged 20 years and older from the 2017–2020 National Health and Nutrition Examination Survey (NHANES). After excluding individuals with frequent alcohol consumption, hepatitis B/C infection, those lacking liver ultrasound examinations, and samples with missing data, a total of 2,460 subjects were ultimately included. The dataset was split into training and testing sets in an 80:20 ratio. Five machine learning algorithms—XGBoost, Random Forest (RF), and Logistic Regression (LR), among others—were utilized to build prediction models, while Recursive Feature Elimination (RFE) was employed to identify key predictive factors.

### Results

Comparison of the five algorithms revealed that the XGBoost algorithm performed the best. Twelve key features were selected through Recursive Feature Elimination (RFE), and the model achieved an AUC of 0.8740 on the testing set, demonstrating

**Data availability statement:** All data are available from the National Health and Nutrition Examination Survey (NHANES) database (https://wwwn.cdc.gov/nchs/nhanes/Default.aspx).

**Funding:** The author(s) received no specific funding for this work.

**Competing interests:** The authors have declared that no competing interests exist.

excellent predictive accuracy and discriminative ability. SHAP plot analysis of the model showed that waist circumference, BMI, and other factors played a pivotal role in the prediction of MASLD.

## Conclusion

The prediction model developed using the XGBoost algorithm and the 12 selected features demonstrates high efficiency and stability in assessing MASLD risk. This model offers innovative technical solutions and data-driven support for the clinical early identification of high-risk populations, with the potential to optimize and refine MASLD prevention and control strategies.

## Introduction

Metabolic dysfunction-associated steatotic liver disease (MASLD), one of the most prevalent chronic liver disorders worldwide, is spreading at an alarming rate. Epidemiological studies indicate that MASLD affects approximately 25% of the global population [1]. Currently, the global prevalence of MASLD is approximately 30.2%, with regional variations of 30.9% in Asia, 16.1% in Australia, 30.2% in Europe, 29% in North America, and 34% in South America [2]. The disease spectrum ranges from simple steatosis to non-alcoholic steatohepatitis (NASH), and may progress to liver fibrosis, cirrhosis, and even hepatocellular carcinoma [3]. More critically, MASLD is closely associated with systemic diseases, including metabolic syndrome, type 2 diabetes (T2D), and cardiovascular diseases, significantly increasing all-cause mortality [4]. Cardiovascular diseases (CVD) are the most common cause of mortality in MASLD patients. MASLD and CVD share several common risk factors including obesity, insulin resistance, and T2D [5]. Importantly, NASH also increases the risk of extra-hepatic complications, especially cardiovascular diseases (CVD), which are among the most common causes of death in NASH patients [6]. Indeed, the alterations in hepatic lipid metabolism that lead to MASLD also drive the development of atherogenic dyslipidemia. Altered glucose metabolism and insulin resistance, also hallmarks of MASLD, can further exacerbate CVD risk in these patients [5]. As a result, MASLD has become a major public health concern that threatens global well-being [7].

Early diagnosis and intervention are critical for improving the prognosis of MASLD patients [8]. During the simple steatosis stage, lifestyle modifications and the management of metabolic risk factors can effectively reverse the disease and prevent its progression to more severe stages [9]. Nevertheless, MASLD remains significantly underdiagnosed, particularly in its early phases, due to several clinical challenges. Firstly, the majority of patients with simple steatosis are asymptomatic and exhibit normal conventional liver enzyme levels, reducing the likelihood of clinical detection. Secondly, there is a lack of standardized, widely accessible non-invasive screening protocols specifically targeting at-risk populations in primary care settings. Moreover, healthcare providers often prioritize managing established metabolic conditions such

as diabetes or hypertension over screening for MASLD, further contributing to underdiagnosis. As a result, many patients are only diagnosed at advanced stages, missing the optimal window for intervention [10]. Traditional diagnostic methods, such as liver biopsy (the gold standard), are invasive, costly, and associated with potential complications, limiting their applicability in large-scale screening [11]. Non-invasive approaches, such as serological markers and imaging examinations, are widely used but suffer from limited sensitivity and specificity for early steatosis, failing to meet the requirements for early and accurate diagnosis [12].

In recent years, machine learning has demonstrated significant potential in disease prediction due to its powerful data processing and pattern recognition capabilities [13,14]. By uncovering hidden associations within large clinical datasets, machine learning algorithms can construct high-precision predictive models for early disease detection [15]. Although several exploratory studies have applied machine learning in MASLD research, existing models still have room for improvement in terms of feature optimization, algorithm selection, and clinical applicability [16]. Therefore, leveraging the large-scale and multi-dimensional data from the National Health and Nutrition Examination Survey (NHANES) [17], this study systematically compares the performance of multiple machine learning algorithms and optimizes the model structure using Recursive Feature Elimination (RFE) [18]. The resulting model offers several key advantages over previous approaches: it relies on easily obtainable clinical indicators, which significantly reduces prediction costs and enhances feasibility for widespread implementation; furthermore, the model improves clinical interpretability by identifying the most influential risk factors. In terms of interpretability, most previous ML models for MASLD (such as traditional deep learning and early tree-based models) often suffer from a "black-box nature," making it difficult to quantify the impact of features on prediction results. This limits clinicians' trust in model outputs and hinders practical application. In this study, SHAP (SHapley Additive exPlanations) analysis was integrated into the XGBoost model. This not only clarifies the predictive contribution of core features (e.g., waist circumference, ALT) but also visually presents the direction of each indicator's impact on disease risk through visualization results (e.g., how increased waist circumference elevates the probability of MASLD). This aligns the model's decision-making logic with clinical understanding of MASLD's pathological mechanisms (e.g., central obesity driving disease development), significantly enhancing clinical acceptability. In terms of accessibility, some previous models rely on advanced imaging features (e.g., MRI fat quantification parameters) or multi-omics data. Such indicators require advanced equipment or complex detection processes, which are difficult to obtain in primary care settings and thus limit practicality. This study only uses routinely collectible indicators, including demographic information, results of physical examinations (e.g., waist circumference), and basic biochemical indicators (e.g., ALT). No additional testing costs or equipment investment are needed, allowing primary care institutions to apply the model directly using routine clinical data. This greatly lowers the threshold for implementation and better meets the clinical needs of large-scale preliminary screening.The goal is to develop an efficient and reliable MASLD prediction model, offering new strategies for the early identification and precise prevention of the disease.

## Methods

### Data source and study design

This study utilized data from the National Health and Nutrition Examination Survey (NHANES) database (2017–2020), a continuous, nationally representative survey program that systematically collects multi-dimensional data from the non-institutionalized U.S. population through questionnaires, physical examinations, and laboratory tests, providing invaluable resources for public health research [19]. For the prediction of Metabolic dysfunction-associated steatotic liver disease (MASLD), strict inclusion and exclusion criteria were applied: initial screening used the liver controlled attenuation parameter (CAP) from transient elastography (FibroScan) to diagnose samples with CAP ≥ 302 dB/m as fatty liver [20]. Subsequent exclusions included frequent drinkers (≥2 times/week), individuals under 20 years of age, hepatitis B/C virus carriers, and those without liver ultrasound results. To address biases induced by missing values, a case-wise deletion

approach was employed, removing any subjects with missing data on key indicators to ensure data integrity. Following this, 2,460 eligible participants remained after screening and data cleaning. Using Python 3.9, the dataset was divided into an 80:20 training set (n = 1,968) and test set (n = 492), with 24 candidate features—including demographics (gender, age, race, education), BMI, and liver function indices—used to construct a high-quality research cohort through strict implementation of the inclusion criteria, thus laying a solid foundation for subsequent model training and validation.

## Features

This study incorporated a wide range of multi-dimensional feature variables to construct the MASLD prediction model. Demographic characteristics included gender (male, female), age, race (Mexican American, other Hispanic, non-Hispanic White, non-Hispanic Black, other races), and educational attainment (Less Than 9th Grade, 9–11th Grade, High School Graduate, Some College or AA degree, College Graduate or above), which help to capture population-based differences in disease susceptibility. Physical measurement indicators included body mass index (BMI, kg/m²), waist circumference (cm), systolic blood pressure (SBP, mmHg), and diastolic blood pressure (DBP, mmHg), which were used to assess obesity and blood pressure status—both of which are significant risk factors for MASLD. Biochemical indicators included liver function markers such as alanine aminotransferase (ALT, IU/L), alkaline phosphatase (ALP, IU/L), and aspartate aminotransferase (AST, IU/L), as well as metabolic-related indices such as blood urea nitrogen (BUN, mmol/L), creatine phosphokinase (CPK, IU/L), and globulin (g/L). These biochemical markers reflect the body's physiological and metabolic status from multiple perspectives and possess potential predictive value for MASLD development.

## ML algorithms

This study utilized five machine learning algorithms [21] to develop MASLD prediction models: Logistic Regression (LR), Random Forest (RF), LightGBM (LGBM), CatBoost, and XGBoost. As a classic linear classification algorithm, LR features a simple structure and strong interpretability, often serving as a benchmark for evaluating the performance of other algorithms [22]. RF reduces variance by integrating multiple decision trees, enhancing model stability and generalization ability, and excels at handling high-dimensional, nonlinear data [23]. LGBM optimizes computational efficiency through histogram-based algorithms, offering low memory consumption, fast training speed, and high prediction accuracy for large-scale datasets [24]. CatBoost is optimized for categorical variables, allowing for efficient processing of large-scale categorical data without the need for complex feature engineering [25]. XGBoost, an optimized implementation of Gradient Boosting Decision Trees (GBDT), supports parallel computing and regularization, demonstrating exceptional performance in handling nonlinear data and complex relationships [26]. The finally selected XGBoost parameters (learning_rate = 0.02, max_depth = 4, min_child_weight = 5, subsample = 0.8, reg_alpha = 2, reg_lambda = 20, etc.) correspond to the highest average validation AUC (0.874).

In the model evaluation phase, the area under the receiver operating characteristic curve (AUC) was used as the primary metric, complemented by a comprehensive analysis of accuracy, sensitivity (recall), specificity, and other performance indicators. To optimize algorithm performance, grid search was applied to all models for hyperparameter tuning, systematically exploring parameter combinations (e.g., learning rate, tree depth, and number of estimators for boosting algorithms) to determine the optimal configuration. After tuning, the XGBoost algorithm achieved an AUC of 0.9020 in the training set and 0.8738 in the test set, significantly outperforming other algorithms and thereby being selected as the core prediction model. Subsequent feature selection using the Recursive Feature Elimination (RFE) algorithm retained 12 key features for the final model, which achieved AUC values of 0.8960 and 0.8740 for the training and test sets, respectively. These results fully validate the model's effectiveness and stability.

## Statistical analyses

This study employed a rigorous statistical analysis approach, using appropriate statistical tests for different data types to ensure scientific validity and reliability. For categorical variables such as gender, race, and educational attainment,

chi-square tests were used to evaluate distribution differences between the MASLD and non-MASLD groups, with results presented as percentages (%) representing the proportion of each categorical feature in the respective groups.

For continuous variables, normality was first assessed using the Shapiro-Wilk test and the Kolmogorov-Smirnov test. If the data conformed to a normal distribution, independent sample t-tests were applied to compare mean differences between groups, with results expressed as mean±standard deviation (Mean±SD). For non-normally distributed data, the Wilcoxon Mann-Whitney U test was used for intergroup comparisons, and data characteristics were reported as median (interquartile range [IQR]). In this study, a two-sided p-value of < 0.05 was set as the threshold for statistical significance, where values below this cutoff indicated statistically significant differences between the groups.

## Results

### Baseline characteristics

A total of 2,460 participants were included in this study, with 715 (29.1%) in the MASLD group and 1,745 (70.9%) in the non-MASLD group. Baseline characteristic analysis revealed significant differences (P<0.05) between the two groups across multiple indicators (Table 1). In terms of demographic features, the MASLD group had a significantly higher proportion of males (51.75%) compared to the non-MASLD group (42.52%, P<0.001) and a greater mean age (median: 55.0 years vs. 49.0 years), with racial distribution also showing statistically significant differences (P<0.001). Regarding educational attainment, only 19.58% of the MASLD group held a bachelor's degree or higher, compared to 25.16% in the non-MASLD group (P=0.031).

For physical measurements and biochemical parameters, the MASLD group exhibited significantly higher values in BMI (median: 34.2 kg/m² vs. 27.2 kg/m²), waist circumference (114.0 cm vs. 94.7 cm), blood pressure, liver function markers (e.g., ALT, GGT), and metabolic indices (e.g., SUA, triglycerides, CRP, GLU) (all P<0.001). Conversely, HDL levels were significantly lower in the MASLD group (1.14 mmol/L vs. 1.37 mmol/L, P<0.001). By contrast, indices such as CPK, LDH, and TBIL showed no significant differences between the groups (P>0.05).These baseline differences indicate that factors such as gender, age, obesity, and metabolic disorders are closely associated with MASLD pathogenesis, providing critical evidence for feature selection and predictive analysis in subsequent machine learning models.

### Performance evaluation of classification models

In the test set evaluation of the Metabolic dysfunction-associated steatotic liver disease (MASLD) prediction model (Fig 1), receiver operating characteristic (ROC) curves were plotted to compare the performance of Logistic Regression (LR), Random Forest (RF), LightGBM (LGBM), CatBoost, and XGBoost models. The results demonstrated that the XGBoost model outperformed the others, achieving an AUC of 0.8738, followed by CatBoost (0.8625), RF (0.8658), LGBM (0.8633), and LR (0.8541). The ROC curves for all models were positioned above the diagonal, indicating predictive capabilities, with XGBoost showing the highest accuracy in distinguishing MASLD cases from non-cases.

Table 2 presents a comprehensive evaluation of five machine learning algorithms—Logistic Regression (LR), Random Forest (RF), LightGBM (LGBM), CatBoost, and XGBoost—using various metrics, including the area under the receiver operating characteristic curve (AUC), accuracy, precision, sensitivity, specificity, and F1 score. The results indicated that the XGBoost algorithm exhibited the best performance in both the training and test sets, with an AUC of 0.9020 in the training set and 0.8738 in the test set. It also outperformed the other algorithms in test set accuracy (0.7988), precision (0.6170), specificity (0.7937), and F1 score (0.7009). In comparison, the test set AUCs for LR, RF, LGBM, and CatBoost ranged from 0.8541 to 0.8658, with some models displaying imbalances between sensitivity and specificity. The XGBoost algorithm's balanced performance across multiple metrics, along with its notable advantages, underscores its superior applicability and reliability for Metabolic dysfunction-associated steatotic liver disease (MASLD) prediction, providing robust support for the development of efficient prediction models.

**Table 1. Baseline characteristics of participants by MASLD status.**

| Parameter | Total (n = 2460) | Without MASLD (n = 1745) | MASLD (n = 715) | p-value |
|---|---|---|---|---|
| Gender | | | | <0.001 |
| Male | 1112 (45.2%) | 742 (42.52%) | 370 (51.75%) | |
| Female | 1348 (54.8%) | 1003 (57.48%) | 345 (48.25%) | |
| Age | 52.0 (36.0, 64.0) | 49.0 (33.0, 63.0) | 55.0 (41.5, 65.0) | <0.001 |
| Race | | | | <0.001 |
| Mexican American | 331 (13.46%) | 205 (11.75%) | 126 (17.62%) | |
| Other Hispanic | 262 (10.65%) | 189 (10.83%) | 73 (10.21%) | |
| Non-Hispanic White | 805 (32.72%) | 545 (31.23%) | 260 (36.36%) | |
| Non-Hispanic Black | 606 (24.63%) | 470 (26.93%) | 136 (19.02%) | |
| Other Race | 456 (18.54%) | 336 (19.26%) | 120 (16.78%) | |
| Education | | | | 0.031 |
| Less Than 9th Grade | 193 (7.85%) | 132 (7.56%) | 61 (8.53%) | |
| 9-11th Grade | 281 (11.42%) | 192 (11.0%) | 89 (12.45%) | |
| High School Grad | 601 (24.43%) | 409 (23.44%) | 192 (26.85%) | |
| Some College or AA degree | 806 (32.76%) | 573 (32.84%) | 233 (32.59%) | |
| College Graduate or above | 579 (23.54%) | 439 (25.16%) | 140 (19.58%) | |
| BMI (kg/m²) | 29.0 (24.9, 34.2) | 27.2 (23.7, 31.3) | 34.2 (29.65, 39.5) | <0.001 |
| Waist (cm) | 99.7 (88.3, 112.52) | 94.7 (84.6, 105.0) | 114.0 (102.9, 124.8) | <0.001 |
| SBP (mmHg) | 121.0 (109.0, 134.25) | 119.0 (108.0, 132.0) | 126.0 (115.0, 139.0) | <0.001 |
| DBP (mmHg) | 74.0 (67.0, 82.0) | 73.0 (66.0, 80.0) | 77.0 (70.0, 85.0) | <0.001 |
| ALT (IU/L) | 17.0 (13.0, 25.0) | 16.0 (12.0, 22.0) | 22.0 (16.0, 32.0) | <0.001 |
| ALP (IU/L) | 75.0 (62.0, 90.25) | 73.0 (60.0, 89.0) | 79.0 (65.0, 95.0) | <0.001 |
| AST (IU/L) | 19.0 (15.0, 23.0) | 18.0 (15.0, 22.0) | 19.0 (16.0, 26.0) | <0.001 |
| BUN (mmol/L) | 5.0 (3.93, 6.07) | 5.0 (3.93, 6.07) | 5.0 (4.28, 6.43) | <0.001 |
| CPK (IU/L) | 110.0 (74.0, 177.0) | 111.0 (74.0, 177.0) | 108.0 (74.0, 175.0) | 0.545 |
| Globulin (g/L) | 31.0 (28.0, 34.0) | 31.0 (28.0, 34.0) | 31.0 (29.0, 34.0) | 0.008 |
| GGT (IU/L) | 20.0 (14.0, 30.0) | 18.0 (13.0, 26.0) | 26.0 (19.0, 38.0) | <0.001 |
| LDH (IU/L) | 154.0 (136.0, 174.0) | 153.0 (135.0, 174.0) | 155.0 (138.0, 175.0) | 0.123 |
| TBIL (umol/L) | 6.84 (5.13, 10.26) | 6.84 (5.13, 10.26) | 6.84 (5.13, 10.26) | 0.093 |
| SUA (umol/L) | 315.2 (261.7, 368.8) | 303.3 (249.8, 356.9) | 339.0 (285.5, 404.5) | <0.001 |
| HDL (mmol/L) | 1.29 (1.09, 1.55) | 1.37 (1.14, 1.63) | 1.14 (0.98, 1.34) | <0.001 |
| Triglyceride (mmol/L) | 1.0 (0.68, 1.49) | 0.88 (0.63, 1.31) | 1.35 (0.95, 1.92) | <0.001 |
| LDL (mmol/L) | 2.74 (2.2, 3.36) | 2.71 (2.2, 3.34) | 2.79 (2.22, 3.39) | 0.316 |
| TC (mmol/L) | 4.63 (4.01, 5.34) | 4.63 (3.98, 5.33) | 4.63 (4.02, 5.35) | 0.502 |
| CRP (mg/L) | 2.02 (0.84, 4.48) | 1.57 (0.71, 3.56) | 3.42 (1.6, 7.32) | <0.001 |
| GLU (mmol/L) | 5.72 (5.33, 6.38) | 5.61 (5.27, 6.05) | 6.33 (5.66, 7.63) | <0.001 |

## Feature selection and final prediction model

Feature selection is a critical step in enhancing model performance and generalization ability for metabolic dysfunction-associated steatotic liver disease (MASLD) prediction. This study initially included 24 candidate features, covering multi-dimensional variables such as gender, age, race, educational attainment, BMI, and liver function indices. To reduce the risk of overfitting caused by feature redundancy, Recursive Feature Elimination (RFE) was applied based on the XGBoost model for feature screening.

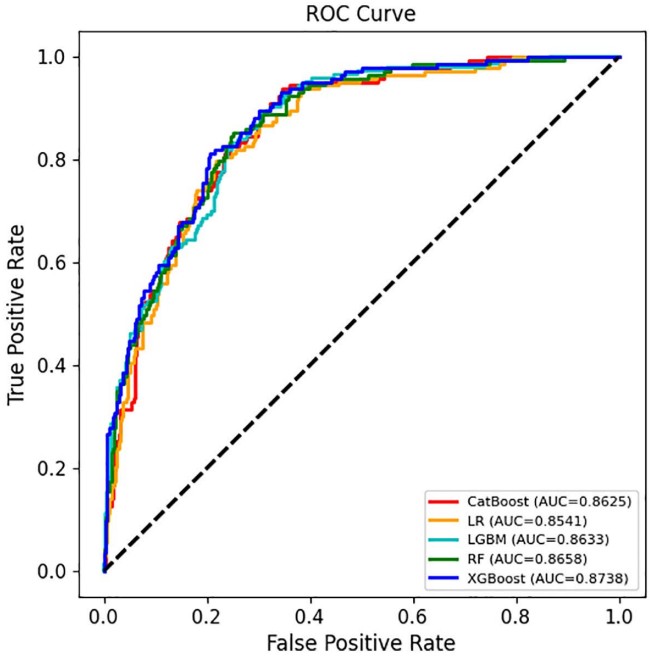

**Fig 1. Comparison of receiver operating characteristic (ROC) curves of five machine learning models in non - alcoholic fatty liver disease (NAFLD) prediction.**

**Table 2. Performance of models by different algorithms.**

| Model | AUC Training | AUC Testing | Accuracy Testing | Precision Testing | Sensitivity Testing | Specificity Testing | F1Score Testing |
|---|---|---|---|---|---|---|---|
| LR | 0.8587 | 0.8541 | 0.7866 | 0.6000 | 0.7972 | 0.7822 | 0.6847 |
| RF | 0.8716 | 0.8658 | 0.7785 | 0.5810 | 0.8531 | 0.7479 | 0.6912 |
| LGBM | 0.8888 | 0.8633 | 0.7744 | 0.5777 | 0.8322 | 0.7507 | 0.6819 |
| CatBoost | 0.8718 | 0.8625 | 0.7358 | 0.5255 | 0.9371 | 0.6533 | 0.6734 |
| XGBoost | 0.9020 | 0.8738 | 0.7988 | 0.6170 | 0.8112 | 0.7937 | 0.7009 |

The RFE algorithm operates iteratively: in each iteration, it removes the feature with the lowest contribution to the model, re-trains the model, and evaluates its performance until the preset number of features or performance criteria are met. After feature screening via RFE (Fig 2), a total of 12 key features were retained, including Age, Race, Body Mass Index (BMI), Waist Circumference, Alanine Transaminase (ALT), Blood Urea Nitrogen (BUN), Globulin, Gamma-Glutamyl Transpeptidase (GGT), High-Density Lipoprotein (HDL), Triglycerides, C-Reactive Protein (CRP), and Glucose (GLU). These features exhibit clear biological significance in MASLD pathogenesis and were significantly associated with MASLD in univariate analysis (P<0.05).

An XGBoost prediction model was reconstructed based on the selected features, and the results showed that the model maintained strong performance on both the training set and the test set (Fig 3), with AUC values of 0.8960 and 0.8740, respectively. Notably, the test set AUC of the optimized model (0.8740) was nearly identical to that of the initial model (0.8738), suggesting that the feature combination effectively reduced model complexity while maintaining high prediction accuracy. The model's AUC is 0.874 with a 95% confidence interval of 0.842–0.906. In addition, the model's accuracy, precision, sensitivity, specificity, and F1 score on the test set were 0.7907, 0.6010, 0.8322, 0.7736, and 0.6979, respectively (Table 3), further confirming the effectiveness of the feature selection and algorithm combination. These

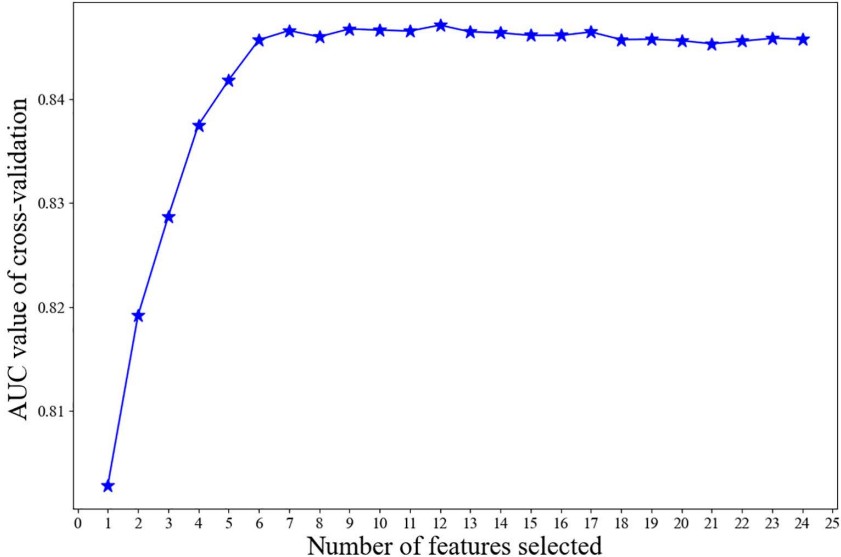

**Fig 2. Performance of the XGBoost model based on 12 selected variables for NAFLD prediction.**

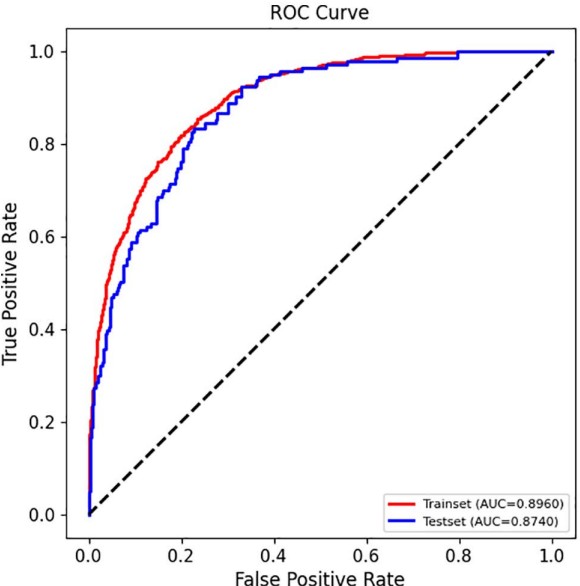

**Fig 3. ROC curves of the training set and the testing set under the XGBoost algorithm.**

results demonstrate that the selected feature combination accurately captures the key risk factors of MASLD, providing robust data support for efficient and reliable disease prediction in clinical practice.

## Post-hoc interpretation results

This study utilized SHAP (SHapley Additive exPlanations) value visualization to analyze in detail the influence of the 12 selected features on the output of the XGBoost prediction model (Fig 4). The results revealed that waist circumference,

**Table 3. Evaluation metrics of the XGBoost model with 12 selected features on training and test sets.**

| Model | AUC Training | AUC Testing | Accuracy Testing | Precision Testing | Sensitivity Testing | Specificity Testing | F1Score Testing |
|-------|-------------|-------------|------------------|-------------------|---------------------|---------------------|-----------------|
| XGBoost | 0.8960 | 0.8740 | 0.7907 | 0.6010 | 0.8322 | 0.7736 | 0.6979 |

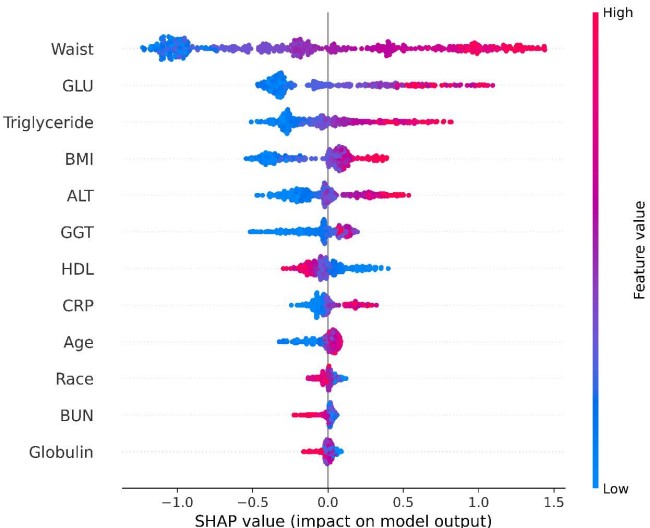

**Fig 4. SHAP Values of 12 Selected Predictors for NAFLD in the XGBoost Final Model.**

blood glucose (GLU), and triglycerides were core features influencing model predictions, as evidenced by their wide distribution ranges and large absolute SHAP values (as shown in the figure). SHAP value analysis demonstrated a significant positive correlation: as waist circumference, blood glucose, and triglyceride levels increased, the predicted probability of MASLD increased substantially. Specifically, high values of waist circumference and triglycerides (represented by the red distribution) were directly associated with higher MASLD prediction probabilities (positive SHAP values). Although blood glucose showed complex interactions in certain samples—where extremely high values occasionally reduced prediction probabilities—the overall trend indicated a dominant positive contribution to MASLD risk.

To intuitively present the effect size, we supplemented the SHAP Bar chart of core features (Fig 5), using "mean absolute SHAP value" to quantify the average impact magnitude of each feature on the model's prediction results. As shown in the figure, waist circumference has the largest mean absolute SHAP value (approximately 0.7), indicating that it has the strongest effect size—consistent with the mechanism that "central obesity is the core driving factor of MASLD". The effect sizes of metabolic indicators such as blood glucose and triglycerides decrease in turn (approximately 0.35 and 0.22, respectively), clearly reflecting hierarchical differences and enhancing clinical interpretability.

## Discussion

This study developed a prediction model for Metabolic dysfunction-associated steatotic liver disease (MASLD) using machine learning algorithms based on data from the National Health and Nutrition Examination Survey (NHANES), with findings of substantial significance for clinical practice and future research.

In model construction, the XGBoost algorithm identified 12 core features via the Recursive Feature Elimination (RFE) method, including metabolic and liver function indicators that align closely with the pathogenesis of MASLD. Among these, metabolic indicators such as waist circumference, triglycerides, and blood glucose had significant impacts in SHAP

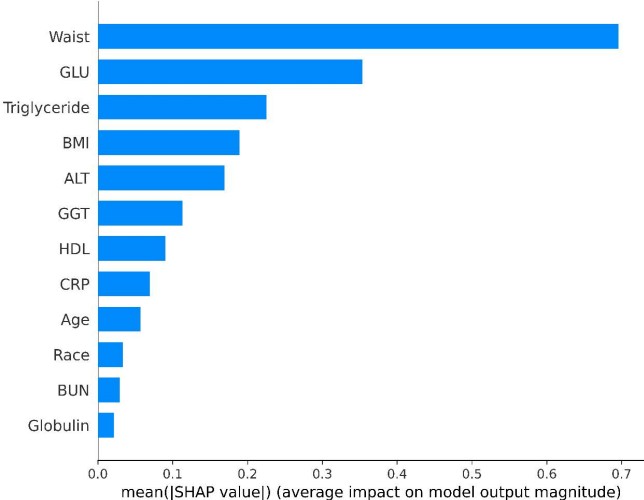

**Fig 5. SHAP Feature Importance: Mean Absolute SHAP Values in the XGBoost Final Model for Predicting MASLD.**

value analysis, highlighting that metabolic disorders are key driving factors for MASLD [27–29]. This aligns with existing knowledge that central obesity, dyslipidemia, and glycometabolic abnormalities accelerate the progression of MASLD by promoting hepatic fat deposition, inflammatory responses, and insulin resistance [30]. The final XGBoost model achieved an AUC of 0.8740 on the test set, significantly outperforming other algorithms. These results indicate that the model effectively captures MASLD risk features and offers a quantitative tool for the early identification of high-risk populations in clinical settings.

From a clinical application perspective, the core features selected in this study are all routine diagnostic indicators that are easily accessible. The prediction model built on these features demonstrates high practicality and generalizability. Clinicians can monitor patients' waist circumference, blood glucose, liver function, and other relevant indicators, and combine these measurements with the model's predictions to implement early interventions for individuals at high risk of MASLD. Such interventions could include dietary adjustments and increased physical activity, which may help delay disease progression and reduce the risk of severe complications, such as cirrhosis and liver cancer [31,32].

However, this study has certain limitations. On one hand, it relies on liver ultrasound for MASLD diagnosis, and the absence of liver biopsy as a pathological gold standard may result in missed diagnoses of some mild-to-moderate cases, potentially affecting the model's accuracy in identifying early-stage disease [33]. On the other hand, the study sample is primarily drawn from the U.S. population, whose genetic background, lifestyle, and environmental exposures exhibit regional specificity. The model's generalizability to other racial or regional populations still requires further validation.

Future research could enhance the model's applicability to different populations by incorporating multi-center and transnational cohort data, alongside multimodal information such as liver elastography and metabolomics. Additionally, long-term follow-up studies could dynamically evaluate the model's efficacy in predicting disease progression, thereby providing stronger support for the precise prevention and control of MASLD.

## Conclusions

This study developed a Metabolic dysfunction-associated steatotic liver disease (MASLD) prediction model using machine learning algorithms based on nationally representative data from the National Health and Nutrition Examination Survey (NHANES), providing an innovative approach for early disease risk identification. Among the five evaluated algorithms— including XGBoost, Random Forest, and Logistic Regression—the XGBoost model demonstrated superior predictive

performance. Through Recursive Feature Elimination (RFE), 12 core predictors were selected from 24 candidate variables, with metabolic indicators (e.g., waist circumference, triglycerides, blood glucose) and liver function parameters (e.g., alanine aminotransferase, gamma-glutamyl transferase) playing pivotal roles in model construction. The model achieved an area under the receiver operating characteristic curve (AUC) of 0.8740 on the test set, significantly outperforming other algorithms and exhibiting robust discriminative and predictive capabilities for MASLD risk. SHAP value analysis further emphasized the dominant influence of metabolic features, such as waist circumference and triglycerides, aligning closely with the metabolic dysfunction-driven pathogenesis of MASLD and offering a quantitative basis for clinical prioritization of key risk factors.

Despite these strengths, the study has several limitations. First, the reliance on liver ultrasound for MASLD diagnosis—without the pathological gold standard of liver biopsy—may lead to missed diagnoses of mild-to-moderate cases. Second, the predominantly U.S.-based sample population limits the generalizability of the findings to other racial or geographic groups. Future research could improve the model's universality and predictive accuracy by incorporating multi-center, cross-national cohort data and integrating multimodal information, such as liver elastography and metabolomics. Such advancements would strengthen global efforts toward early MASLD prevention and control by providing more robust technical support.

## Supporting information

**S1 Data.**
(ZIP)

## Author contributions

**Conceptualization:** Yong Zhang, Xiang Liu, Xingqiang Zhang, Xiaoxu Li.

**Data curation:** Yong Zhang, Xiang Liu, Xingqiang Zhang, Xiaoxu Li.

**Formal analysis:** Yong Zhang, Xiang Liu, Xingqiang Zhang.

**Funding acquisition:** Yong Zhang, Xiang Liu, Xingqiang Zhang.

**Investigation:** Yong Zhang, Xiang Liu, Xingqiang Zhang.

**Methodology:** Yong Zhang, Xiang Liu, Xingqiang Zhang.

**Project administration:** Yong Zhang, Xiang Liu, Xingqiang Zhang.

**Resources:** Yong Zhang, Xiang Liu, Yangfan Fei.

**Software:** Yong Zhang, Xiang Liu, Yangfan Fei.

**Supervision:** Yong Zhang, Xiang Liu, Yangfan Fei.

**Validation:** Yong Zhang, Xiang Liu, Yangfan Fei.

**Visualization:** Yong Zhang, Xiang Liu, Yangfan Fei.

**Writing – original draft:** Yong Zhang, Xiang Liu, Xiaoxu Li.

**Writing – review & editing:** Xiang Liu, Xiaoxu Li.

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
