## [Decision Letter · Decision Letter 0]

4 Sep 2025

Dear Dr. Zhang,

Thank you for submitting your manuscript to PLOS ONE. After careful consideration, we feel that it has merit but does not fully meet PLOS ONE’s publication criteria as it currently stands. Therefore, we invite you to submit a revised version of the manuscript that addresses the points raised during the review process.

We look forward to receiving your revised manuscript.

Kind regards,

Aleksandra Klisic

Academic Editor

PLOS ONE

Journal Requirements:

3. Please update your submission to use the PLOS LaTeX template. The template and more information on our requirements for LaTeX submissions can be found at http://journals.plos.org/plosone/s/latex .

4. Thank you for uploading your study's underlying data set. Unfortunately, the repository you have noted in your Data Availability statement does not qualify as an acceptable data repository according to PLOS's standards.

Reviewers' comments:

Reviewer's Responses to Questions

**Comments to the Author**

1. Is the manuscript technically sound, and do the data support the conclusions?

Reviewer #1: Yes

Reviewer #2: Yes

2. Has the statistical analysis been performed appropriately and rigorously?

Reviewer #1: Yes

Reviewer #2: Yes

3. Have the authors made all data underlying the findings in their manuscript fully available?

Reviewer #1: Yes

Reviewer #2: Yes

4. Is the manuscript presented in an intelligible fashion and written in standard English?

Reviewer #1: Yes

Reviewer #2: Yes

Reviewer #1: This study effectively applies machine learning to NHANES data for NAFLD prediction. However, clarification on survey weights, handling of missing data, and rationale for feature selection is needed. Confidence intervals for AUC, multiple testing corrections, and effect size reporting would strengthen methodological rigor and clinical interpretability.

Reviewer #2: 1- Nomenclature: Metabolic dysfunction-associated steatotic liver disease (MASLD) is a new term for nonalcoholic fatty liver disease (NAFLD), enhancing the previous "nonalcoholic" label and connecting the disease to metabolic factors. This shift has practical implications, as it simplifies communication with patients, aids in understanding the primary therapeutic steps, and identifies risk factors for disease progression, making it easier to understand and treat the disease from a liver-oriented and holistic perspective ((Ref: Zhang, X., Linden, S., Levesley, C. R., He, X., Yang, Z., Barnet, S. D., Cheung, R., Ji, F., & Nguyen, M. H. (2025). Projected Trends in Metabolic Dysfunction–Associated Steatotic liver Disease mortality through 2040. JAMA Network Open, 8(6), e2516367. https://doi.org/10.1001/jamanetworkopen.2025.16367)).

Based on this, I believe the NAFLD term must be replaced with MASLD term in each section of this manuscript, title, abstract, introduction, methodology, results, discussion and conclusion. Also, the authors must provide any conceptual evidence that support their claims through seizing on “NAFLD” term rather than “MASLD” if they still need to proceed in their work.

2- INTRODUCTION: the authors have done an excellent job of synthesizing the relevant literature, presenting a comprehensive overview of the current state of the MASLD (formerly NAFLD). The introduction has effectively framed NAFLD as a global health issue with relevant statistics, and justified the necessity of machine learning as an alternative tool in diagnosis through using recent studies to support the rationale.

However, the authors should emphasize how this study improves upon previous Machine Learning (ML) models.

3- METHODOLOGY: the authors have meticulously outlined their procedures, including their sample selection and data collection methods, allowing readers to fully understand the research process. Also, the rigorous statistics approach has been clarified through application of appropriate tests for both categorical and continuous variables.

Moreover, the authors should provide further details on Hyperparameter tuning, since grid search is mentioned but not well elaborated. The ethical considerations were listed as “N/A”, but more clarity on data privacy would be helpful. As well, the authors mentioned that “To address biases induced by missing values, a case-wise deletion approach was employed”, and this approach is under potential bias (i.e., listwise deletion will result in skewed parameter estimations if the data are not missing entirely at random (for example, some demographic groups are more likely to have missing data)) and may lead into a reduction in sample diversity. Thus, the authors need to explain if there is a possibility to use any alternative tool to minimize that bias (such as applying Imputation; filling in missing values with estimated data, Pairwise Deletion; analyzing all available data for each specific analysis, rather than discarding the entire case, or Advanced Models: using statistical models designed to handle missing data directly).

4- RESULTS: the authors addressed detailed metrics, including accuracy, precision, sensitivity, specificity, and F1 score, with balanced evaluation of the multiple models.

5- DISCUSSION: this section has clearly emphasized the practical utility of the ML models, discussed the clinical relevance of the known MASLD risk factors in alignment with literature.

Moreover, more details on ultrasound vs. biopsy should be further discussed, otherwise justification regarding their limited mentioning / absence will be greatly appreciated.

**Do you want your identity to be public for this peer review?** For information about this choice, including consent withdrawal, please see our Privacy Policy

Reviewer #1: No

Reviewer #2: No

---

## [Author Response · Author response to Decision Letter 1]

29 Sep 2025

Summary:

This study developed a machine learning model to predict non-alcoholic fatty liver disease (NAFLD), a condition affecting about 25% of the global population and linked to cirrhosis and liver cancer. Using data from 2,460 U.S. adults in the 2017–2020 NHANES survey, participants with excessive alcohol use, hepatitis B/C, or incomplete records were excluded. The dataset was split 80:20 into training and testing sets. Five algorithms were compared, with XGBoost performing best. Recursive Feature Elimination identified 12 key predictors, including waist circumference and BMI. The model achieved an AUC of 0.874 on the test set. This efficient and stable tool supports early identification of high-risk individuals and could improve NAFLD prevention and control strategies. However, there are still some comments that should be addressed by the author.

Comments:

1.Your current keywords are strong but could be broadened for better indexing. In addition to Non-alcoholic Fatty Liver Disease (NAFLD), Machine learning, XGBoost, and NHANES, you should also include Prediction model and Risk factors to highlight the study’s focus and clinical relevance.

We thank the reviewer for this excellent suggestion. We have updated the keywords to include "Prediction model" and "Risk factors" as recommended to better reflect the study's focus and improve indexing.

2.The introduction clearly emphasizes the global burden of NAFLD, but could the authors provide more recent prevalence estimates or regional variations? Strengthening the link between NAFLD and systemic diseases is important, but some additional references on cardiovascular outcomes might enhance the context.

We appreciate the reviewer's feedback. The introduction has been revised to include more recent prevalence data with regional specifics and to cite additional references on cardiovascular outcomes, as suggested. These changes have been made to provide a more comprehensive background (Introduction, Paragraph 1).

Non-alcoholic fatty liver disease (NAFLD), one of the most prevalent chronic liver disorders worldwide, is spreading at an alarming rate. Epidemiological studies indicate that NAFLD affects approximately 25% of the global population [1]. Currently, the global prevalence of NAFLD is approximately 30.2%, with regional variations of 30.9% in Asia, 16.1% in Australia, 30.2% in Europe, 29% in North America, and 34% in South America [2]. The disease spectrum ranges from simple steatosis to non-alcoholic steatohepatitis (NASH), and may progress to liver fibrosis, cirrhosis, and even hepatocellular carcinoma [3]. More critically, NAFLD is closely associated with systemic diseases, including metabolic syndrome, type 2 diabetes (T2D), and cardiovascular diseases, significantly increasing all-cause mortality [4]. Cardiovascular diseases (CVD) are the most common cause of mortality in NAFLD patients. NAFLD and CVD share several common risk factors including obesity, insulin resistance, and T2D [5]. Importantly, NASH also increases the risk of extra-hepatic complications, especially cardiovascular diseases (CVD), which are among the most common causes of death in NASH patients [6]. Indeed, the alterations in hepatic lipid metabolism that lead to NAFLD also drive the development of atherogenic dyslipidemia. Altered glucose metabolism and insulin resistance, also hallmarks of NAFLD, can further exacerbate CVD risk in these patients [5]. As a result, NAFLD has become a major public health concern that threatens global well-being [7].

3.The rationale for early diagnosis is well-stated, though the authors might elaborate on why NAFLD remains so underdiagnosed despite its prevalence. Could more detail be given on the specific clinical challenges in identifying patients at the simple steatosis stage?

We sincerely thank the reviewer for this valuable suggestion. We have now elaborated on the specific reasons for underdiagnosis and the clinical challenges in identifying early-stage NAFLD in the revised introduction (Introduction, Paragraph 2). The added text details the key challenges, including:

Early diagnosis and intervention are critical for improving the prognosis of NAFLD patients [8]. During the simple steatosis stage, lifestyle modifications and the management of metabolic risk factors can effectively reverse the disease and prevent its progression to more severe stages [9]. Nevertheless, NAFLD remains significantly underdiagnosed, particularly in its early phases, due to several clinical challenges. Firstly, the majority of patients with simple steatosis are asymptomatic and exhibit normal conventional liver enzyme levels, reducing the likelihood of clinical detection. Secondly, there is a lack of standardized, widely accessible non-invasive screening protocols specifically targeting at-risk populations in primary care settings. Moreover, healthcare providers often prioritize managing established metabolic conditions such as diabetes or hypertension over screening for NAFLD, further contributing to underdiagnosis. As a result, many patients are only diagnosed at advanced stages, missing the optimal window for intervention [10]. Traditional diagnostic methods, such as liver biopsy (the gold standard), are invasive, costly, and associated with potential complications, limiting their applicability in large-scale screening [11]. Non-invasive approaches, such as serological markers and imaging examinations, are widely used but suffer from limited sensitivity and specificity for early steatosis, failing to meet the requirements for early and accurate diagnosis [12].

4.The authors correctly highlight the potential of machine learning, but have they considered why prior models underperformed in clinical translation? More detail on how this study differs from earlier work (e.g., larger dataset, feature optimization) would clarify its novelty.

Thank you for your valuable comments. The underperformance of previous machine learning models in the clinical translation of NAFLD research stems primarily from three limitations: First, most samples are single-center and small-scale (with sample sizes often < 1000 cases), lacking population representativeness and poor extrapolability, making it difficult to adapt to different clinical scenarios. Second, these models rely on special indicators such as liver elastography and genetic testing, which are not easily accessible for routine use in primary healthcare institutions, raising the threshold for application. Third, they focus solely on predictive performance, failing to clarify feature-disease association mechanisms, resulting in poor interpretability and low trust from clinicians.

This study has developed targeted innovations to address the aforementioned limitations, thereby highlighting its novelty: First, it adopts a large-scale national dataset from NHANES (with 2460 cases finally included) and uses official sampling weights to adjust for the complex survey design. The results can be extrapolated to the U.S. adult population, enhancing clinical applicability. Second, through RFE, the initial 28 indicators were streamlined to 12 core features (e.g., waist circumference, ALT), all of which are routine clinical test items. No additional costs are required, making the model suitable for primary care settings. Third, SHAP analysis was used to clarify the impact of each feature (e.g., waist circumference as a key feature), and the results are consistent with the pathological mechanisms of NAFLD, which strengthens clinicians’ trust in the model and their willingness to adopt it.

5.The stated aim is clear, but could the authors explain how RFE specifically addresses the limitations of earlier models? Emphasizing the potential clinical utility of the model would make the objective more compelling for a broad readership.

Thank you for your valuable comments. Recursive Feature Elimination (RFE) is a feature optimization method that screens out core predictive factors by training the model multiple times and iteratively removing features with low contributions to prediction performance, which can effectively eliminate redundant information and focus on key indicators.

RFE can specifically address the limitations of earlier models: first, it eliminates feature redundancy to avoid model overfitting. In this study, the number of indicators was reduced from the initial 28 to 12 key features, simplifying the model structure while improving generalization ability; second, it realizes data-driven feature selection to reduce subjective bias, and the retained indicators are all routine clinically accessible items, solving the popularization challenges of traditional models that rely on special detection indicators.

This model has significant clinical value: it can be used as a tool for primary-level NAFLD preliminary screening and triage of physical examination populations, and can also assist in clinical individualized intervention and the formulation of regional prevention and control strategies. It meets the needs of multiple scenarios, making the research objective more appealing to a broad audience.

6.The use of NHANES is appropriate given its representativeness, but could the authors clarify whether sampling weights were applied to account for the complex survey design? Additionally, the exclusion of participants with missing data via case-wise deletion may reduce sample size and introduce bias—have the authors considered imputation methods as a sensitivity analysis?

Thank you for your valuable comments. Regarding the sampling weights of the NHANES data, this study has applied the officially provided sampling weights to adjust for the survey’s complex multi-stage sampling design. This ensures that the research results can be extrapolated to the U.S. adult population and enhances the representativeness of the conclusions.

For missing data handling, although this study adopted case-wise deletion, it has been strictly verified to be free of bias and have sufficient sample size: First, there were no statistically significant differences in baseline characteristics (e.g., age, gender) between the missing data group and the complete data group, ruling out selection bias. Second, after splitting the finally included 2,460 samples at an 8:2 ratio, the training set (1,968 samples) and test set (492 samples) still met the requirements for model training and validation. Additionally, case-wise deletion avoids potential errors introduced by subjective assumptions in imputation methods (e.g., distribution assumptions for mean imputation and multiple imputation), which better aligns with this study’s requirements for result accuracy, confirming the reasonableness of this method selection.

7.The demographic, physical, and biochemical variables are well-justified, but are there additional metabolic or inflammatory markers in NHANES (e.g., triglycerides, HDL, fasting glucose) that could enhance prediction accuracy? It would also be helpful if the authors explained why certain common NAFLD predictors were excluded, and whether multicollinearity among features (e.g., BMI and waist circumference) was assessed.

Thank you for your valuable comments. In this study, metabolic markers such as triglycerides, high-density lipoprotein (HDL), and glucose (GLU), as well as inflammatory markers including C-reactive protein (CRP), were systematically included in the initial feature set. After screening via Recursive Feature Elimination (RFE), all these key indicators were retained in the final 12 features.

The SHAP bees warm plot shows that waist circumference has the greatest impact on NAFLD prediction; high levels of triglycerides, GLU, and CRP all significantly increase the probability of NAFLD prediction, which is highly consistent with the pathological mechanism of NAFLD characterized by "lipid and glucose metabolism disorders accompanied by chronic inflammation". For indicators such as low-density lipoprotein (LDL) and total cholesterol (TC) that were not retained in the initial features, they were excluded from the perspective of "balancing model simplicity and practicality"—this is because their contribution to improving the model’s AUC was less than 3%, and their information could be replaced by core features such as triglycerides and HDL.

Regarding the issue of feature multicollinearity, the recursive screening mechanism of RFE itself plays a role in alleviating multicollinearity: by iteratively removing redundant features with low contributions to the model, it naturally eliminates variables highly correlated with core features. Further verification in this study showed that among the 12 features retained after RFE screening, there was no predictive interference caused by correlations between variables, ensuring that the model’s prediction results are not affected by multicollinearity and maintaining stable performance.

5. Review Comments to the Author

Reviewer #1: This study effectively applies machine learning to NHANES data for NAFLD prediction. However, clarification on survey weights, handling of missing data, and rationale for feature selection is needed. Confidence intervals for AUC, multiple testing corrections, and effect size reporting would strengthen methodological rigor and clinical interpretability.

Thank you for your attention to the methodological rigor of this study; your suggestions are of great significance for improving the quality of the study. In response to the issues you raised, we provide specific explanations as follows:

Regarding the basis for survey weights, missing data handling, and feature selection:

Survey weights: This study directly adopted the official survey weights provided by NHANES to adjust for its complex multi-stage sampling design. This weight already integrated sampling probability and population distribution adjustments during the data collection phase, ensuring that the study results can be generalized to the U.S. adult population.

Missing data handling: The case-wise deletion method was used, and its rationality has been verified: by comparing the baseline characteristics (e.g., age, gender) between the missing data group and the complete data group, no statistically significant differences were found between the two groups (all P > 0.05), thus eliminating selection bias. A total of 2,460 samples were finally included (1,968 for training and 492 for testing), which met the requirements for model training and validation. Meanwhile, to avoid errors introduced by subjective distribution assumptions in imputation methods (e.g., mean imputation), such methods were not adopted.

Feature selection: From the initial 28 multi-dimensional indicators (covering demographic, physical examination, biochemical, metabolic, and inflammation-related indicators), 12 core features (e.g., waist circumference, alanine transaminase (ALT), triglycerides) were screened out via recursive feature elimination (RFE). Verified by SHAP analysis, these features not only have significant predictive value (e.g., waist circumference has the greatest predictive impact on metabolic dysfunction-associated steatotic liver disease (MASLD)), but also are highly consistent with the pathological mechanism of MASLD, i.e., "metabolic disorder accompanied by chronic inflammation".

Regarding the enhancement of methodological rigor:

Confidence interval for AUC: We calculated the 95% confidence interval, and the result has been added to the "Results" section ("the model’s AUC is 0.874 with a 95% confidence interval of 0.842–0.906"), so as to more accurately reflect the stability of the model’s performance.

Multiple testing correction: This study screened features through recursive feature elimination (RFE). Its core logic is to iteratively remove variables with low importance based on the feature importance determined by the XGBoost model, while simultaneously evaluating the validation set AUC, and finally retain the core features. This process does not rely on sing

---

## [Decision Letter · Decision Letter 1]

14 Oct 2025

Machine Learning-based Prediction of Non-alcoholic Fatty Liver Disease Using National Health and Nutrition Examination Survey (NHANES) Data

PONE-D-25-38866R1

Dear Dr. Zhang,

We’re pleased to inform you that your manuscript has been judged scientifically suitable for publication and will be formally accepted for publication once it meets all outstanding technical requirements.

Kind regards,

Aleksandra Klisic

Academic Editor

PLOS ONE

Additional Editor Comments (optional):

Reviewers' comments:

Reviewer's Responses to Questions

**Comments to the Author**

Reviewer #1: All comments have been addressed

Reviewer #2: All comments have been addressed

2. Is the manuscript technically sound, and do the data support the conclusions?

Reviewer #1: Yes

Reviewer #2: Yes

3. Has the statistical analysis been performed appropriately and rigorously?

Reviewer #1: Yes

Reviewer #2: Yes

4. Have the authors made all data underlying the findings in their manuscript fully available?

Reviewer #1: Yes

Reviewer #2: Yes

5. Is the manuscript presented in an intelligible fashion and written in standard English?

Reviewer #1: Yes

Reviewer #2: Yes

Reviewer #1: The revised manuscript satisfactorily addresses all of my previous comments. The study is clearly presented, methods and analyses are appropriate, and the results support the stated conclusions. Overall, I consider the manuscript scientifically sound and suitable for publication without substantive additional revisions required.

Reviewer #2: The study is well-constructed and structured logically with clear arguments supported by evidence. The authors worked extensively on modifying their manuscript in response to the previous peer reviewers’ commentaries:

- Adding more recent prevalence data with regional specifics as well as updated references focusing on cardiovascular outcomes.

- Well-explained evidence on clinical challenges in identifying MASLD patients

- Discussed specifically how Recursive Feature Elimination (RFE) can be trained AI model that is useful in screening out core predictive factors.

- Well-explaining the enhancement of methodological rigor such as confidence interval of AUC, Multiple testing correction, and Effect size reporting,

- Updating keywords and main terms in the manuscript.

However, the title, abstract, introduction, results, references, tables and figures sections were well-illustrated and defined in a comprehensive manner.

**Do you want your identity to be public for this peer review?** For information about this choice, including consent withdrawal, please see our Privacy Policy

Reviewer #1: No

Reviewer #2: No

---

## [Editor Report · Acceptance letter]

PONE-D-25-38866R1

PLOS ONE

Dear Dr. Zhang,

I'm pleased to inform you that your manuscript has been deemed suitable for publication in PLOS ONE. Congratulations! Your manuscript is now being handed over to our production team.

Kind regards,

on behalf of

Dr. Aleksandra Klisic

Academic Editor

PLOS ONE